# The Community Pharmacy as a Study Center for the Epidemiological Analysis of the Population Vaccination against SARS-CoV-2: Evaluation of Vaccine Safety and Pharmaceutical Service

**DOI:** 10.3390/pharmacy12010016

**Published:** 2024-01-20

**Authors:** Jacopo Raffaele Dibenedetto, Michela Cetrone, Marina Antonacci, Domenico Pio Cannone, Stefania Antonacci, Pasquale Bratta, Francesco Leonetti, Domenico Tricarico

**Affiliations:** 1Management Control Unit, Puglia Regional PHT Office, Pharmaceutical Territorial Area (ASL BA), Via L. Starita, 6, 70132 Bari, Italy; j.dibenedetto@studenti.uniba.it (J.R.D.); michela.cetrone@asl.bari.it (M.C.); marina.antonacci@uniba.it (M.A.); stefania.antonacci@asl.bari.it (S.A.); pasquale.bratta@asl.bari.it (P.B.); 2Department of Pharmacy-Pharmaceutical Science, University of Bari Aldo Moro, Via Orabona 4, 70125 Bari, Italy; francesco.leonetti@uniba.it; 3Pharmacy Dr. Cannone Domenico Pio, Via Alvisi, 40, 76121 Barletta, Italy; cannoned@farmaciacannone.191.it

**Keywords:** vaccine, SARS-CoV-2, distribution/dispensation of behalf of third parties, D.P.C., antivirals, prescriptions, cardiovascular reactions, observational study, community pharmacy, C.PHARM

## Abstract

We conducted a monocentric observational study aimed at evaluating the vaccine safety and the pharmaceutical service provided at a community pharmacy (C.PHARM) in the Puglia Region in the period from 29 December 2021 to 12 March 2022 using data from 550 patients of various ages and sexes and with concomitant diseases. We collected anamnestic data, the number of hospitalizations, and any post-vaccination adverse reactions. Interviews using the integrated EQ5 method were also performed to evaluate the quality of the service offered and any therapy preference. As expected, the vaccines were reactogenic after the first dose in the patients with mild–moderate reactions, with younger age and female gender as risk factors. Immune-allergic reactions of a moderate–severe degree were observed in adult females. In the elderly, the vaccination was well tolerated. Comirnaty^®^ showed a favorable O.R. < 1 vs. other vaccines. No cardiovascular events or hospitalizations were observed up to May 2023. Regional data indicate that all treatments during May 2023 were correlated with the viremia. Paxlovid^TM^ was prescribed in 3% of the patients in our center and in 1.46% in the region, and distributed/dispensed on behalf of third parties in accordance with a novel distribution/dispensation protocol of the C.PHARM that resulted in a safe vaccination center providing appropriate patient inclusion during vaccination.

## 1. Introduction

On 11 January 2020, ten days after the first notification made to the World Health Organization (WHO) by China of the first cases of pneumonia in Wuhan, the first gene sequence of the SARS-CoV-2 virus was published [1]. This made it possible to start the development of vaccines against the virus, and scientists, industries, and other organizations around the world have collaborated to rapidly develop safe and effective vaccines. Some vaccines have been created through methods already known before the pandemic, while others have been developed using innovative technologies already employed in tumors but with varying success [2], or approaches used during previous emergencies such as the SARS and Ebola pandemics in West Africa.

The vaccines, when administered, simulate the first contact with the infectious agent, stimulating an immunological response (humoral component and a cellular component) like a natural infection without causing the disease and the complications derived to acquire active immunity. This is due to immunological memory, or the ability of the immune system to remember which foreign microorganisms have attacked the body in the past and to respond in a timely manner. The absence of immunological memory is the reason why children contract infectious diseases more frequently than adults; in fact, without vaccinations, the human body can take weeks to produce enough antibodies to counteract a microorganism. There are two general vaccination strategies [3], one universal or massive, and the other selective, with the aim of protecting certain population groups with a particularly high-risk rate of disease, such as the elderly, pregnant women, immunosuppressed individuals or individuals who have more opportunities to infect others (e.g., health workers). When we talk about immunization programs, we must not only think about the health of the individual, but also about reducing the social, psychological, and economic repercussions of the disease on individuals and on the health system. In fact, a fundamental prerogative, such as that of the anti-SARS-CoV-2 vaccination plan, is a reduction in the pressure on the health and social assistance system, an area in which a community pharmacy can play an essential role. Vaccines can lead to individual protection and contribute to community protection. For a sufficiently sized population, in many cases, more than 80% of the resident population are immune to an infectious disease. It is unlikely that the disease can spread. This allows people who cannot be vaccinated (people allergic to the components of the vaccine) to benefit from others being vaccinated, as the disease does not spread easily in the community, which is known as herd immunity. Herd immunity is not achieved with all vaccines; in fact, if a vaccine confers individual protection from a disease but does not prevent the spread of the infectious agent, failure to vaccinate the subject is considered a risk for the individual only, and not the community, such as in the case of the tetanus vaccine. Often, a single dose is not enough. For each vaccine, an optimal number of doses is established according to the age and condition of the subject, and everything is reported on a technical sheet. If the number of doses is more than one, the recommended interval between doses and the minimum or maximum time between doses is indicated. In the case of SARS-CoV-2, the protection does not last a lifetime, and booster doses are indicated [4]. As with all medicines, however, vaccines are not 100% effective; in fact, the effectiveness for everyone depends on a series of factors such as age, health status, previous or subsequent contact with the pathogen, the mode of administration, and the vaccine itself. Vaccines are not effective immediately after the first administration, but the effectiveness increases to an acceptable level in 14 days, an effect linked to the development of specific immunoglobulins.

In the case of SARS-CoV-2, mRNA vaccines were often used in the development strategy [5]. This type of vaccine represents an innovation in that instead of providing the antigen, they provide the genetic information necessary to synthesize and express the antigen by host cells through messenger RNA (mRNA) or self-replicating RNA. This technology, despite being newly introduced, has been studied for more than a decade in cancer [2,6]; Dr. Katalin Karikó and Dr. Drew Weissman won the Nobel Prize for Physiology and Medicine 2023 for their discoveries concerning nucleoside base modifications that enabled the development of effective mRNA vaccines against COVID-19. The mRNA contains a genetic sequence with instructions to produce the identified target protein spike, a protein found on the surface of the SARS-CoV-2 virus. The mRNA has the characteristic of being able to be inactivated very easily, and is not able to enter a cell on its own. For this reason, the mRNA of the vaccine has been inserted into lipid nanoparticles, which has a protective function and allows them to enter the cells of the host. After administration, the mRNA contained within the nanoparticles enters the host cells and is read by ribosomes, and the spike protein is synthesized. At this point, the spike protein is transported to the surface of the cell. The presence of this protein, foreign to the host, will stimulate the immune system. Specifically, it leads to the activation of T lymphocytes and the production of antibodies that prevent the SARS-CoV-2 virus from entering cells. It is important to note two aspects of mRNA vaccines. The first is that mRNA does not enter the nuclei of cells and therefore cannot interact with DNA. The second is that because the vaccine does not contain the virus, only the genetic information needed to synthesize the spike protein, it cannot cause the disease. There are technologies that use plasmid DNA to make host cells code the spike protein. Vaccines that use this technology often have the great advantage of being able to be produced on a large scale, and have high stability. Unfortunately, however, they generally show low immunogenicity and need to be administered through special devices. In contrast, recombinant protein vaccines use viral proteins to induce an antigenic response. The proteins that have been used to produce vaccines are the spike protein, the receptor binding protein (RBD), and virus-like particles (VLPs), such as that in Novavax’s Novaxovid vaccine.

The authorized vaccines are the mRNA vaccines Comirnaty (Pfizer—BioNTech, Mainz, Germany) and Spikevax (Moderna—NIAID, Bethesda, MD, USA), the DNA vaccine Vaxzevria (University of Oxford—AstraZeneca, Gaithersburg, MD, USA), and Janssen’s COVID-19 vaccine (Johnson & Johnson, New Brunswick, NJ, USA). They were firstly used in the vaccination campaigns in different European Member States. Their available formulations, such as Comirnaty Original/Omicron BA.1, Comirnaty Original/Omicron BA.4–5, Spikevax Bivalent Original/Omicron BA.1, and the Spikevax Bivalent Original/Omicron BA.4–5, are effective against variants [7]. Currently, seven vaccines for COVID-19 have been approved by the European Medicinal Agency (EMA, Amsterdam, the Netherlands) and authorized for marketing release in the E.U. In addition, the Nuvaxovid (Novavax) vaccine was approved in December 2021, while the COVID-19 inactivated and adjuvanted Valneva vaccine (Valneva Austria GmbH, Vienna, Austria), and VidPrevtyn Beta (Sanofi Pasteur, Abingdon, VA, USA), were recently released and are available for use. Those vaccines are often based on the original strain of the SARS-CoV-2 virus, while VidPrevtyn Beta is based on the Beta variant. Also, the CoronaVac (Sinovac, Beijing, China) COVID-19 vaccine is an inactivated virus vaccine that has been approved for emergency use by the World Health Organization (WHO) [8], and the Sinopharm Beijing Institute of Biological Products COVID-19 vaccine (BBIBP-CorV) has also received authorization.

The development of the COVID-19 vaccines has been particularly rapid [9]; the alignment of the preclinical and clinical development phases of vaccines has been an important element, as has the continuous and facilitated dialogue between pharmaceutical companies and authorities as a result of the EMA Pandemic Task Force (COVID-ETF) made up of experts with various skills [10,11]. In the case of COVID-19 vaccines, the Conditional Marketing Authorization was adopted [12,13] by the EMA and the Italian Agency of Drugs (AIFA) authorities based on the positive benefit/risk balance of the medicinal product evaluated by the Committee for Medicinal Products for Human Use (CHMP). The Conditional Marketing Authorization differs substantially from Emergency Use Authorization, which allows some countries such as the United States and Great Britain to temporarily use unauthorized medicines in emergency conditions [12,13]; this authorization does not correspond to a marketing authorization and could, if necessary, be withdrawn at any time.

In this context, vaccine vigilance is of interest. The agencies periodically review new safety information for all available vaccines and analyze the data through vaccine vigilance [14], monitoring whether or not the benefit/risk ratio (B/R) remains favorable over time during the vaccine campaign. Vaccine vigilance is carried out with different modalities, such as passive and active surveillance. Passive surveillance consists essentially of the analysis of so-called spontaneous reports and case reports; at least three consistent case reports are needed in the absence of any confounding factors. Active surveillance, on the other hand, means a set of proactive actions regarding the stimulation and collection of reports at sentinel sites, such as groups of patients or doctors with intensive monitoring systems, and through registers consisting of lists of patients with the same characteristics of pathology or exposure to drugs or vaccines. For the new COVID-19 vaccines, the methods for the storage, handling, and preparation of the dose could generate errors and possibly adverse reactions that must be reported by pharmacists. Through the analysis of spontaneous reports, signs can emerge relating to risks that need to be quantified. For this purpose, post-marketing cohort and case–control studies are mainly used, which may involve many patients and aim to verify the efficacy and safety of vaccines in the authorized therapeutic indications and under the real conditions of use. These studies allow us to identify infrequent and rare adverse reactions that did not occur during pre-registration studies conducted on small and homogeneous samples of patients, and therefore to study subpopulations of subjects not previously evaluated. The Strategic Plan for SARS-CoV-2 vaccination drawn up by all the institutions involved in the management of the vaccination campaign (Ministry of Health, ISS, AIFA, etc.) allows the AIFA to promote additional activities with respect to passive pharmacovigilance, essentially represented by active pharmacovigilance and pharmaco-epidemiology studies similar to that which we propose here in our work. Active pharmacovigilance includes all the activities of stimulation and the collection of reports on the populations of vaccinated subjects that are registered in the vaccination registers, and the pharmaco-epidemiology includes all observational studies on cohorts or case–control studies on the population of those vaccinated or special populations (i.e., with specific conditions). The objective of these activities is to raise safety signals for regulatory purposes, improve knowledge on new vaccines in general, and adequately communicate the real risks of vaccination.

To date, in the case of COVID-19 vaccines [15], mild and moderate adverse reactions to the two innovative mRNA vaccines have been reported to be very similar. Local reactions are among the most common, and generally consist of pain in the arm or redness at the site where the dose was administered. In some cases, some small swelling may also occur, or axillary lymphadenopathy may be associated. However, these are mild reactions that resolve spontaneously within a short time. Among the most frequent systemic reactions are fever, chills, fatigue, headache, and muscle pain, which are observed with biologics due to immunoreactions and inflammatory responses [16,17,18]. More rarely, gastrointestinal symptoms, such as nausea, vomiting and diarrhea, have been reported. Myocarditis and pericarditis need further study [19]. It has already been reported that they are rare events of a severe degree; the EMA estimates that they occur in one to two cases per one hundred thousand people vaccinated, with a slightly higher incidence after the administration of Spikevax^®^ compared to Comirnaty^®^. They generally occur in young males and mostly after the second dose. Symptoms occur within 14 days of vaccination and consist of palpitations or feelings of the heart in the throat, difficulty breathing, and chest pain. As for the course of the disease, it is resolved with rest or treatment. During the vaccination campaign, other rare serious adverse reactions were also recorded, but they are not related to the Spikevax^®^ and Comirnaty^®^ vaccines; this is the case, for example, of thrombosis with thrombocytopenic syndrome, which has been associated with vaccination with viral vector vaccines, and mainly occurred in women under 50 years of age [20]. Guillain–Barré syndrome (GBS), on the other hand, is a rare neuropathy that leads to muscle weakness or even paralysis for longer or shorter periods of time.

Regarding vaccination in children, the data collected to date show that using a reduced dosage of Comirnaty^®^ in the age group of 5–11 years and a standard dosage in the age group of 12–17 years is safe as well as effective. Adverse reactions reported to the Pharmacovigilance System in the United States have shown that adverse reactions in this age group are similar to those seen in adults. In a real-world cohort, serious COVID-19 vaccine adverse effects were rare, and comparisons across brands could be made, revealing that full vaccination dose, vaccine brand, younger age, female sex, and having had COVID-19 before vaccination were associated with greater odds ratios of adverse effects [15,21].

Therefore, three emerging safety signals were identified by the COVID-19 subcommittee: transverse myelitis, hearing loss and tinnitus, and acute hepatitis. In addition, the long-term impact of myocarditis, pregnancy outcomes, and GBS are being continuously monitored, as recommended by the subcommittee [14], but new signals can emerge in the long term.

The communication between the pharmacist and the patients plays a role in the success of the vaccine campaign to combat vaccine hesitancy and risk perceptions that affect patient preference. This is part of a social and cultural context that is affected both by the immediate effects of the pandemic experience and by the profound changes that in the last 30 years have characterized the approach to prevention and health promotion and the collective perception of vaccinations. Various causes have been associated with reluctance and uncertainty in resorting to vaccination in COVID-19 [22]. The concerns associated with COVID-19 vaccines are linked to a number of considerations that can fuel vaccine hesitancy and reduce trust in vaccines (virus news, conflicting messages, rapid vaccine development, perceived politicization of the process, distrust in traditional health information sources, spread of online misinformation, and conflicting public debate on vaccines) and confirm the need for planned and coordinated communication that ensures the rapid dissemination of coherent messages, the construction of collaborations, and the paying of attention to the different levels of health of the population, and in particular to equity, which can prevent vaccine hesitancy and promote individual confidence in the vaccine. Vaccine hesitancy origin can also be ethnicity-based [23,24], and strategies can be applied to minimize it [25]. Several risk-related characteristics are known to systematically influence people’s perception. For instance, the risks associated with pandemic vaccination, primarily in adults, were more accepted during the first year of anti-COVID-19 vaccination activities, but with the extension of vaccination coverage, first in the age group between 12 and 18 years and subsequently in the age group of 5–12 years, these risks were progressively perceived as much less acceptable. The risks related to complications of COVID-19 after not being vaccinated (i.e., the consequence of not having to make any decisions) are more accepted than the risks resulting from the proactive decision to get vaccinated. The research pointed out that very often there is a discrepancy between the subjective perception of risk and objective hazard assessment [26]. For instance, the risk can be elevated in populations affected by intellectual disability [27], and it can be neurologically based [28]. The risk of thrombosis is higher in these patients than in the general population, despite the extensive use of novel direct anticoagulants [29]. According to the international guidelines, low-molecular-weight heparin is recommended first after the careful evaluation of the side effects related to using the drug, and it is necessary to evaluate bleeding risk as a universal strategy for routine thrombosis prevention using standard-dose unfractionated heparin or low-molecular-weight heparin in COVID-19 patients admitted to general hospitals other than ICUs [30]. In a minority of cases, mid-dose low-molecular-weight heparin could also be considered [31]. In addition, a high-risk population is represented by the patients undergoing opioid treatments that, during the pandemic, received take-home therapy for addiction, with an enhanced risk of overdoses [32]. In this complex scenario, it is essential to consider some essential elements to develop effective communication strategies for COVID-19 vaccines, and the local pharmacist and the family doctor play a role in translating the messages directed to a large population into a patient-personalized message to reduce vaccine hesitancy [23,24]; vaccine campaigns are indeed often not personalized.

Pharmacists, as health professionals, already play a strategic role in strengthening healthcare through, for instance, the reconciliation of therapy [33,34]. The role of pharmacists is expressly considered in the National Plan of Chronicity [35], which specifically provides for the full involvement of pharmacies in health education and primary and secondary prevention activities through the provision of innovative professional services. There is talk of a new pharmacy model that, in addition to medicine, provides citizens with a series of additional services. In fact, even during the COVID-19 emergency, the pharmacy carried out important assistance activities involving testing for the presence of IgG and IgM antibodies and the execution of rapid antigenic swabs for the detection of the SARS-CoV-2 antigen.

Berlofa Visacri and coworkers examined trials involving pharmacists during the pandemic and their roles. The 11 studies included in this review were conducted in the United States of America (n = 4), China (n = 4), Saudi Arabia (n = 1), Taiwan (n = 1), and Macau (n = 1). Most studies described the work of pharmacists in hospitals (n = 8), in ambulatory pharmacies (n = 4), community pharmacies (n = 2), and clinics (n = 1). Participants in the included studies were varied, including healthcare professionals (n = 7), COVID-19 patients (n = 5), patients in general hospitalization (n = 2), the general population (n = 2), pediatric patients (n = 1), solid organ transplant patients (n = 1), patients on warfarin therapy (n = 1), and patients with myelofibrosis (n = 1). In these studies, the various activities of the pharmacists were demonstrated, including disease prevention and infection control. These include the distribution of medical devices, the development of hygiene strategies, pharmaceutical counseling on drugs dispensed to COVID-19 patients and not used, the proper storage and supply of the drug (e.g., a drug formulary for the treatment of COVID-19), guiding the supply and purchase of medications, the conversion of intravenous to oral medication administration when possible, and patient care and support for healthcare professionals (e.g., ensuring appropriate use of the drug for patients and healthcare professionals). All studies made one-to-one contact with recipients, and in six studies they used group contact. Several methods of communication have been reported, including face-to-face (n = 4), written (n = 5), telephone (n = 6), video conferencing (n = 5), and radio stations. Studies were conducted in different intervention settings, such as at the hospital bed (n = 7), the hospital pharmacy (n = 2), the community pharmacy (n = 2), the outpatient setting (n = 4) and the home of the recipient (n = 5).

More recently, the successful administration of casirivimab/imdevimab in an outpatient setting with a low rate of adverse events was performed in a U.S. community pharmacy. This innovative monoclonal antibody administration service should be used as an example of a call to action for the expansion of pharmacists’ scope of practice. The role of the pharmacist in the pediatric influenza and pediatric COVID-19 vaccines has been well supported by evidence [36].

The role of the hospital pharmacist has also emerged; they provided essential support to pediatric patients during the pandemic in an Italian setting in the absence of evidence [37], and in consulting for appropriate prescriptions during pregnancy [38]. Strong emotional exhaustion and stress in the study sample of community pharmacists in Italy has been reported, as well as high risk perceptions and fear. These pharmacists provided an essential service, despite the high risk of infection [39].

Considering the situation in the United States and many countries in the European Union, the possibility of involvement of community pharmacies in administering vaccination via pharmacists is emerging. The Italian National Government, based on specific regulations such as in art. 20 (D.L. n. 41/2021), has allowed, on an experimental basis and limited to 2021, the administration of vaccines against SARS-CoV-2 in community pharmacies, but with specific obligations, such as the certification of the pharmacist by the National Institute Superior of Health (ISS).

We therefore set up a novel vaccination protocol based on the art. 20 [40] (D.L. n. 41/2021) regulatory rules for patients in our center. We have conducted a monocentric retrospective observational study aimed at evaluating the vaccine safety and the pharmaceutical service provided at a local community pharmacy. We collected anamnestic data and data on post-vaccination adverse reactions, as well as interviews using the integrated EQ5 method about the evaluation of quality of life and therapy preference. We evaluated the effectiveness in reducing severe forms of the disease, hospitalizations, and intensive care unit permanence in our center. The data were collected in the period from 29 December 2021 to 12 March 2022, and refer to a sample of 550 patients of various ages and sexes with concomitant diseases and related drug therapy. A follow-up period of 15 months was applied to 100 patients.

The regional consumptions of vaccines, antivirals, and viremia were also analyzed in this work to monitor the prescription, drug distribution/dispensation to patients using the novel D.P.C. protocol, and its correlation with patients’ preference. Vaccine effectiveness was evaluated in terms of the number of hospitalizations in the entire regional population.

## 2. Materials and Methods

### 2.1. Protocol

Patients that attended the vaccination center at the community pharmacy of the ASL BT (Pharmacy Dr. Cannone Domenico Pio) agreed to be subjected to the vaccination cycle prescribed to them according to the methods indicated, according to which the Italian Government [41], based on specific rules, has allowed vaccination on an experimental basis and limited to 2021 at the community pharmacy. To fulfill the framework agreement between the government, regions, autonomous provinces, FederFarma and Assofarm, the community pharmacy in Barletta has used considerable economic resources to set up a room in accordance with the law. The measures for air exchange have been strengthened through a system with no recirculation and a constantly running air extractor. They involve different areas:Acceptance area = booking verification, informed consent collection, pre-vaccination triage;Preparation area = preparation of the solution to be injected;Administration area = vaccine administration with an appropriate emergency cart and a standard container for waste disposal;Monitoring area = space dedicated to the hosting of the vaccinated subject for the surveillance of any adverse reactions for a time ranging from 15 min to 60 min.

The vaccines Comirnaty^®^ and Spikevax^®^ were ordered through the “Valore” portal, and, upon arrival at the pharmacy, batches were provided for the acceptance of the bubbles on the “GoC” portal. Subsequently, the vials were stored in refrigerators intended for the storage of pharmaceutical products with constant monitoring of the temperature set according to the information acquired from the technical data sheets of each vaccine. Before proceeding with the setup, the vials were brought to room temperature for about 15–30 min. The vial was then inspected for particulate matter or discoloration of the vaccine. Subsequently, after disinfecting the shelf with alcohol and covering it with a sterile cloth, the pharmacist proceeded to prepare 1 mL syringes with a 23 G needle with low dead space, which are suitable for intramuscular (i.m.) administration. It is important to underline that the Comirnaty^®^ vaccine provides for a necessary preventive dilution with 1.8 mL of 0.9% NaCl in a 3 mL syringe, while for the Moderna vaccine, no dilution is required, and therefore the amount necessary for administration is taken directly from the vial. In addition, the vials were properly rotated or turned upside down, but without ever shaking them in a whirling manner.

Assisted persons were then invited to sit in the monitoring area for a time of no less than 15 min. The pharmacist (Figure 1) at the end of each vaccination session proceeded to disinfect the surface, mark the vial used by deleting the label with a marker indicating the date and time of start and end of use, and clean the remaining solution or empty vial with ethyl alcohol. Subsequently, all the data required to ensure the updating of the vaccination registry and the traceability of the vaccine were entered into the “Java” computer system, also entering the batch number. Upon completion of the registration operations, the vaccination certificate was printed.

Before proceeding with the execution of the vaccination act, it is necessary to acquire an informed consent form. Informed consent refers to the acceptance that the user expresses of a health treatment, freely and after having been adequately informed about the methods of execution, benefits, side effects, foreseeable risks, any alternative choices, and possible consequences deriving from refusal.

Informed consent must meet the following requirements. ”Informed” means that the information must be understandable, expressed in clear and plain language without using complex scientific acronyms or terms, and possibly in a language well known to the user, based on validated scientific sources, specifying the nature and location of the process, the probability of success, the method of execution, the expected consequences and their methods of resolution, possible complications, possible alternative choices, the consequences of refusal, and subsequent behaviors consistent with the treatment. It must be “conscious”, that is, expressed by a subject capable of understanding and agreeing after being given correct information. In terms of “personnel”, informed consent must be issued by those who will be vaccinated (for adults), by those who exercise parental authority in the case of minors, and by the guardian/legal representative/support administrator for those who are incapable of understanding and willing. It must be collected in an “explicit” way, referring to each given vaccination, and always acquired before the vaccination act. It must also be revocable at any time by the user, and free, that is, collected without any form of conditioning of the person’s will.

An accurate medical history is essential before the administration of any vaccine to identify any contraindications or precautions to be taken. Health personnel, with targeted questions, using a standardized anamnestic card, gave an effective pre-vaccination screening. The purposes of pre-vaccination anamnesis are to identify any conditions that indicate precaution or constitute a contraindication to vaccination, establish good communication with the subject/family, and facilitate conscious adherence to vaccination.

The staff who carry out the anamnesis must be specifically trained regarding precautions, incompatibilities, and true and false contraindications of each vaccination, and must have mastery of the materials and tools provided (anamnestic card, guide to contraindications, etc.). An observation period of at least 15 min is set out after administration of the vaccine. This time interval is established considering that most fast-onset adverse events requiring health intervention begin within 10 min. The observation period should be extended to 60 min in the case of a history of severe allergies to substances not present in vaccines (food, drugs, etc.). In the case of immediate allergies (within 4 h) to the specific vaccine or its constituents, a specialist evaluation must be requested in advance. Vaccination clinics must be equipped with the pharmacological and instrumental equipment necessary for immediate interventions, in adequate and functioning quantities. Rapid onset events after vaccination include the following: respiratory spasms (affective or hiccups), anxiety crises, fainting/collapse, episodes of hypotonia/hypo-responsiveness (HHE), and manifestations of immediate hypersensitivity.

In adults or older children, the most common adverse event is syncope or vasovagal crisis, which occurs immediately after the injection or shortly after. During this episode, the subject quickly pales, loses consciousness, and, if standing or sitting, falls to the ground. The recovery of consciousness takes place in one or two minutes. Syncope is sometimes accompanied by brief clonic shocks; however, if this remains an isolated episode, it does not require any specific treatment or special investigation. Instead, more attention should be directed towards immediate hypersensitivity reactions. These manifestations of hypersensitivity can be mild when symptoms are limited to a urticarial rash and/or immediate swelling at the injection site, or severe when cardiorespiratory and neurological symptoms predominate or when shock occurs with severe hypoperfusion due to relative hypovolemia, with or without bronchospasm and/or laryngospasm or glottis edema. All vaccination recipients should be prepared to deal with an anaphylactic reaction and to distinguish it from syncope, anxiety crises, convulsions, prolonged apnea attacks, or other condition. All vaccination points, including those at our center, are equipped with an emergency cart with drugs ready for use in the case of an immediate hypersensitivity episode.

Generally, vaccination may be recommended in the presence of a precaution, when the benefit to be derived from administering the vaccine outweighs the risk of an adverse reaction. A contraindication is a condition in the recipient that increases the risk of a previously observed serious adverse reaction and contraindicates the administration, whereas a precaution is a condition in the recipient that may increase the risk of a serious adverse reaction or that may impair the ability of the vaccine to produce immunity, and therefore requires a risk/benefit assessment. From a regulatory point of view, the contraindication is a negative effect observed in clinical trials, while the precaution is based on the exclusion criteria of clinical trials (unstudied populations). At the time of vaccination, healthcare professionals performing a vaccination must check for contraindications and/or precautions for use before administering a vaccine through the vaccination history or a collection of information through a series of precise and simple questions, using a standardized card. Some of these may suggest postponing vaccination, for example, in the case of a severe or moderate acute illness with fever or no fever at the time of vaccination. In other cases, such as generalized urticaria occurring immediately after the administration of a previous dose or a severe allergic reaction to latex for products containing natural rubber latex in the prefilled syringe, the benefits and risks of vaccination should be weighed up. In the case of a possible increased risk of allergic reactions, it is possible to organize vaccination in a protected environment, i.e., in vaccination centers (unlike local pharmacies) where maximum assistance is available to deal with a possible anaphylactic reaction. There are contraindications to vaccination that are not supported by evidence. Among the most frequent are allergies in family members, allergies to penicillin, milk proteins or other substances not contained in vaccines, fever after a previous dose of the vaccine, non-serious allergic reactions after the previous dose, previous febrile seizures in family members or those being vaccinated (precaution for measles, mumps, rubella, chickenpox), recent exposure to an infectious disease, mild acute illness without fever or with fever < 38.5 °C, and chronic diseases that do not have specific contraindications, such as diabetes.

The Institute Superior of Health (ISS) has recently updated the indications for vaccination in pregnancy, recommending the extension of the vaccination offer, with mRNA vaccines, to all pregnant women in the second and third trimesters. This recommendation stems from growing evidence of the safety of vaccination in pregnancy, both to the fetus and the mother. In fact, it is believed that the risk of abortion or problems for the fetus or newborn is like that found in unvaccinated women, and in any case is not higher than the probability of experiencing complications, even severe, if the infection were to be contracted during pregnancy. In addition, a woman who is vaccinated during pregnancy could transfer part of the antibodies to the fetus, via the placenta. As for the first trimester of pregnancy, vaccination is not recommended in Italy, as the evidence is still scarce. Therefore, in this period of gestation, it is necessary to carefully assess risks and benefits. During lactation, women can get vaccinated without any interruption. A breastfeeding woman must be informed that vaccination does not expose the infant to risks, and that it is possible that there will be a transfer of antibodies through breast milk.

On the basis of the above considerations, the eligibility criteria of the patients enrolled are as follows:

Adult male and female patients following informed and written consent;Male and female patients aged >65 years old following informed and written consent;Pediatric patients following specific informed and written consent by family members.

The exclusion criteria were as follows:

Urticaria that occurred immediately after administration of a previous dose, or a severe allergic reaction to latex for products containing natural rubber latex in the pre-filled syringe.

### 2.2. Statistics

The data were collected in the period from 29 December 2021 to 12 March 2022, and refer to a sample of 550 patients of various ages and sexes with concomitant diseases and related drug therapy. Anamnestic data and data on post-vaccination adverse reactions were collected on an electronic C.R.F. (Excel Microsoft 10.00) for subsequent graphic and statistical processing under anonymity [32,42,43,44,45] according to the guidelines indicated by Directives 679 and 680 of 2016 and the EU General Data Protection Regulation (GDPR) [46,47]. The person responsible for data privacy of the ASL BA was the lawyer Elisabetta Fortunato (privacy@sanita.puglia.it). Interviews using the EQ-5D-Y method [48] about the quality of the service offered and possible preferences between vaccination choices and drug therapies were performed (Appendix A) for 100 patients. A follow-up period of 15 months was applied to these patients.

The odds ratio (O.R.) for each treatment was calculated vs. other treatments = (N ADR exposed/N exposed)/(1 − (N ADR exposed/N exposed)/(N ADR not exposed/N not exposed)/(1 − (N ADR not exposed/not exposed)).

### 2.3. Vaccine Products and Drugs

Comirnaty (Pfizer—BioNTech) is an mRNA vaccine that, at the beginning of the vaccination campaign, was only authorized for people aged 16 years and older. It is currently also authorized for the age groups 12–15 years and 5–11 years. After diluting the contents of the vial, it is administered intramuscularly (i.m.) into the deltoid region of the arm. In adults, the dose is 0.3 mL in the formulation of 30 micrograms of mRNA per dose, while in children between 5 and 11 years, the dose is 0.2 mL in the formulation of 10 micrograms of mRNA per dose. As mentioned earlier, the primary cycle consists of two doses that are administered 21 days apart. In some clinical conditions of immunocompromised (e.g., transplant) patients, the administration of an additional dose of vaccine is recommended, starting 28 days after the second dose. In addition, in all individuals aged 18 years and older, a booster dose of 0.3 mL is recommended at least 4 months, or 120 days, after completion of the primary course of vaccination [49].

Spikevax (Moderna—NIAID) was authorized by the AIFA in Italy on 7 January 2021, and is authorized from 12 years of age. This is an mRNA vaccine. It is administered intramuscularly in the deltoid region of the arm. The dose for the primary cycle is 0.5 mL, or 100 micrograms per dose. As mentioned above, a primary cycle consists of 2 (two) doses that are administered 28 days apart. In the case of clinical conditions leading to immune impairment, an additional dose may be administered 28 days after the second dose. As with Comirnaty^®^, people aged 18 years and over can administer a booster dose at least 4 months, or 120 days, from the completion of the primary cycle. In this case, it is administered at half the dosage, i.e., 0.25 mL, containing 50 micrograms of mRNA [50].

Vaxzevria (University of Oxford—AstraZeneca) is given via two injections of 0.5 mL into the muscle. The second injection may be given 4 to 12 weeks after the first injection. Vaxzevria^®^ is not recommended for children under 18 years of age. There is currently insufficient information available on the use of Vaxzevria^®^ in children and adolescents under 18 years of age. A preferential use of the AstraZeneca vaccine is that in subjects between 18 and 65 years, for which more solid evidence is available. Prot. 31355 of 15 March 2021 ordered a temporary national ban on the use of the “COVID-19 Vaccine AstraZeneca”, AIC no. 049314026, holder AIC AstraZeneca AB, represented in Italy by AstraZeneca S.p.A., after the signal of thromboembolic events. However, the European Medicines Agency (EMA) lifted, with immediate effect, the prohibition of use on 19 March 2021, as the benefits of the vaccine in preventing hospitalization and death from COVID-19 outweigh the risk of developing disseminated intravascular coagulation or clots in the vessels that drain blood from the brain.

The EMA’s Safety Committee concluded its preliminary review of a signal for thrombi outbreaks in people vaccinated with the AstraZeneca COVID-19 vaccine. The Committee confirmed that the benefits of the vaccine in combating the still widespread threat of COVID-19 (which in turn causes clotting problems and can be fatal) continue to outweigh the risks of side effects, the vaccine is not associated with an increased overall risk of thromboembolic events in those who receive it, and there is no evidence of problems related to specific batches of the vaccine or to particular production sites; however, the vaccine may be associated with very rare cases of thrombi in the presence of thrombocytopenia, i.e., low levels of platelets (blood elements that promote clotting) with or without bleeding, including rare cases of thrombi in the vessels that drain blood from the brain, as in the cerebral thrombosis of the venous sinuses (CVST). These are rare cases; around 20 million people in the UK and European Economic Area (EEA) have received the vaccine as of 16 March 2021, and the EMA has found only 7 cases of thrombi in multiple blood vessels (as seen in disseminated intravascular coagulation (CID)) and 18 cases of CVST. A causal link with the vaccine has not been proven, but it is possible and deserves further analysis. The Pharmacovigilance Risk Assessment Committee (PRAC) involved experts in blood diseases in its review, and worked closely with other health authorities. Overall, the number of thromboembolic events reported post-vaccination, both in pre-authorization studies and in post-vaccination reporting (469 reports, 191 of which are from the EEA), is lower than expected in the general population. This has allowed the PRAC to confirm that there is no increased overall risk of thrombi. However, some concerns remain in younger patients, particularly related to these rare cases.

The Committee’s experts examined in detail the cases of CID and CVST reported by European Member States, nine of which were fatal. Most of these occurred in people under the age of 55, mostly women. Because these events are rare, and COVID-19 itself often causes blood clotting disorders in patients, it is difficult to estimate the expected incidence of these events in people who have not received the vaccine. However, based on pre-COVID-19 data, it was calculated that, as of 16 March 2021, less than one case of CID was expected to occur among people under 50 years of age within 14 days of vaccine administration. This was not evident in the older population given the vaccine. In conclusion, the AIFA, with a circular on 7 April 2021, confirmed that the Vaxzevria^®^ vaccine is approved from 18 years of age, based on current evidence, considering the low risk of thromboembolic adverse reactions in the face of high mortality from COVID-19 in the most advanced age groups, and recommended its preferential use in people over the age of 60. By virtue of the data available to date, those who have already received a first dose of the Vaxzevria^®^ vaccine can complete the vaccination cycle with the same vaccine [51].

Janssen’s COVID-19 vaccine (Johnson & Johnson), which is indicated for active immunization in the prevention of novel coronavirus disease (COVID-19) caused by the SARS-CoV-2 virus in individuals aged 18 years and older, is given as a single 0.5 mL dose via i.m. injection only, preferably into the deltoid muscle of the arm. Following administration of the Janssen COVID-19 vaccine, blood clots have been observed very rarely in association with low platelet levels. This condition included severe cases with blood clots even at unusual sites (e.g., brain, intestines, liver, spleen), in some cases with the presence of bleeding. The cases occurred in the three weeks following vaccination, and mainly in women under 60 years of age. This condition also resulted in death. The CTS met on 20 April 2021 for an update on the discussion and conclusions of the PRAC-EMA on the evaluation of the safety signal related to cases of thrombosis with thrombocytopenia occurring after the administration of the Janssen COVID-19 vaccine. The EMA reiterates that the benefits of the vaccine in the prevention of COVID-19 outweigh the risks of side effects throughout the population included in the authorized indication (subjects from 18 years of age). A combination of thrombosis and thrombocytopenia, in some cases accompanied by bleeding, has been observed very rarely following vaccination with the COVID-19 Janssen vaccine. For this reason, the same conditions of use as applied to the Vaxzevria^®^ vaccine are recommended for the Janssen vaccine. Therefore, the Janssen vaccine, which is approved for use from 18 years of age, should preferably be given to people over 60 years of age [52].

Finally, the Nuvaxovid (Novavax) vaccine was authorized by the AIFA in Italy at the end of December 2021, and is authorized in people from 18 years of age. The development platform is different from those of previous vaccines in that it is a vaccine based on a recombinant protein, the spike protein, and an adjuvant. Doses are 0.5 mL each and contain 5 micrograms of spike protein per dose [53].

#### Antiviral Drug and D.P.C. Protocol

Paxlovid (nirmatrelvir/ritonavir) [54], Pfizer Europe MA EEIG, is used for treating COVID-19 in adults who do not require supplemental oxygen and who are at increased risk of the disease becoming severe. Nirmatrelvir is a protease reversible inhibitor of coronavirus 3CLpro, mainly metabolized by cytochrome P450 (CYP)3A4, and ritonavir, an inhibitor of the CYP3A isoforms that potentiate nirmatrelvir by fixing its suboptimal pharmacokinetic properties. Patients take two nirmatrelvir 150 mg tablets and one ritonavir 100 mg tablet twice daily for five days, starting within five days of symptoms showing. Paxlovid^TM^ is expected to reduce the risk of hospital admission and death in unvaccinated people at risk of severe disease from 7% to 0.8%, although, with the omicron variant, this benefit was reduced. 

This drug was provided to patients using the novel D.P.C. protocol that allows the acquisition of drugs by ASL-BA that are in stock by following a centralized procedure related to improved conditions and their distribution by community pharmacies that dispense the drugs within 24 h to patients at EUR 7 per patient for the service. This drug is available at 50 thousand doses per month.

### 2.4. Vaccine Effectiveness

The effectiveness in our setting was evaluated in relation to different outcomes. We evaluated the efficacy in reducing severe forms of the disease, hospitalizations, and intensive care unit permanence and mortality related to COVID-19 in our patients. In addition, although to a lesser extent, the ability to reduce the number of infections (symptomatic and asymptomatic) of SARS-CoV-2 was considered.

In the report of 7 December 2021, efficacy was estimated from data from the period of 5 July to 5 December 2021. The report shows how COVID-19 vaccines are extremely useful in preventing disease outcomes: the efficacy that is estimated is 88.7% against hospitalizations, 93.5% for intensive care unit (ICU) admissions, and 89.2% against deaths. As expected, the impact on SARS-CoV-2 diagnoses, i.e., for all infections, symptomatic and asymptomatic, is more modest, but still close to 65%. It should be noted that the percentages shown are derived from an estimate, and are therefore subject to margins of uncertainty. For subjects who have completed the primary cycle for more than 5 months (150 days), the effectiveness in preventing severe forms of the disease (which include hospitalizations, ICU admissions, and deaths) is reduced by about 9%, from 92.6% to 83.7%. The administration of an additional dose, or a booster dose, has the effect of bringing the efficacy back above 93%. More marked is the effect of time on the ability to prevent diagnoses of SARS-CoV-2, or the set of asymptomatic and symptomatic infections. In this case, the effectiveness of vaccination, for those who have completed the vaccination cycle for more than 5 months, is reduced by more than 30%, from 74.3% to 39.6%. Even in this case, however, the administration of an additional dose, or a booster dose, brings the vaccination efficacy above 76%. These data are consistent with estimates obtained from surveillance in other countries, and highlight the need to administer the booster dose from a minimum of 4 months to a maximum of 6 months from the second dose.

### 2.5. Vaccinovigilance

Suspected vaccine reactions are reported electronically by filling out the reporting form, available on both the AIFA and Vigifarmaco websites, following the guided procedure. For new COVID-19 vaccines, the same card currently in use for all other vaccines and the current reporting method are used. To define the temporal relationship between vaccination and adverse events, and to identify any therapeutic errors or defects in vaccine quality, we collected information on the date and time of vaccination; type, name, and batch number of the vaccine; site; mode and route of administration; and dose number (first, second, etc.). The report form is sent to the Local Pharmacovigilance Manager (RLFV) responsible for the territory who validates the report and inserts it into the National Pharmacovigilance Network (RNF). The Local Pharmacovigilance Manager responsible for the territory shall validate the reporting form within 7 days of receipt. Validation means the verification of the consistency of the data and the presence of minimum reporting requirements (a patient, a medicine, a reaction, a signaler). Once included in the network, the report is subjected to further quality control by the Regional Pharmacovigilance Centers and AIFA, who verify the completeness of the information on the patient and the vaccine, possibly request further follow-up information, and evaluate the causal link. From the RNF, reports pass daily through the European database Eudravigilance and then to the global database of the WHO, VigiBase.

An adverse event is any undesirable medical event that occurs in a patient or in a subject included in a clinical trial who is given a medicinal product that does not necessarily have a causal relationship with the treatment itself. Adverse drug reaction (ADR), on the other hand, implies that there is a causal relationship between a medicine and the vaccine, and an adverse event/effect is possible.

When an adverse event is reported, the data are analyzed to determine whether there is a causal link with the vaccine. Adverse events could also be due to illness or anxiety about receiving the vaccine.

The safety concepts also used here were as follows: “danger” represents an objective event that can harm people, while “risk” represents the potential damage of variable severity determined by exposure to danger, so the risk expresses a possibility (probability) of damage in relation to exposure to the danger and can be measured in objective terms.

## 3. Results

In our center, a total number of 550 patients composed of adult patients (N patients = 418), aged patients (N patients = 73), and pediatric patients (N patients = 59) were vaccinated on 12 March 2022. The data were analyzed retrospectively and reported according to age groups. Vaccination administration was well tolerated in all population groups in our center. The vaccines were reactogenic after the first dose in >75% of the patients with mild and moderate injection site reactions. The two most administered vaccines were Comirnaty^®^ (N patients = 371) and Spikevax^®^ (N patients = 81). The adult sample was homogeneous in relation to gender for Comirnaty^®^, with an imbalance towards the male sex in the sample treated with Spikevax^®^. The distributions of concomitant diseases in the two treated samples were comparable for tumors, metabolic diseases, allergic diseases and intolerance, but differed in relation to the arthro-muscular diseases, which were most represented in the sample treated with Spikevax^®^, while cardiovascular diseases were more represented in the sample treated with Comirnaty (Figure 2).

The incidence of moderate immune-allergic reactions was approximately double in the sample treated with Spikevax^®^ vs. the sample treated with Comirnaty^®^, and affected the female sex (Figure 2). In detail, in adult patients, vaccination administration caused mild and moderate injection site reactions in all patients with expected tolerance, and in some patients adverse reactions that did not require hospitalization or medical intervention occurred within 24–48 h and were reported in the data sheet.

Uncommon reactions of a moderate degree with an incidence of 1.39% and 2.86% in the adult female patients were observed with Comirnaty^®^ and Spikevax^®^, respectively (Table 1). Specifically, one case of skin rash in a 23-year-old woman not affected by diseases, a case of high fever with a T > 40.5 °C in a 22-year-old woman who did not respond to paracetamol in the absence of diseases, a case of peripheral edema in a 36-year-old woman with a history of food intolerance and chemicals, and one case of syncope in a 35-year-old female patient with hypertension, were recorded. These reactions did not require medical intervention or hospitalization, and were resolved (Table 2).

We did not observe moderate or severe reactions with other vaccines due to the rarity of the event and the low sample size. The five patients vaccinated with Janssen for the first and second doses were then switched to Comirnaty^®^ (N patients = 3) and Spikevax^®^ (N patients = 2) for the booster dose, without showing intolerance or adverse reactions. The eight patients vaccinated with Vaxzevria^®^ were switched for their second (booster) dose to Comirnaty^®^ (N patients = 2) or a booster dose of Spikevax^®^ (N patients = 4) without consequences or intolerance. We did not observe any significant reactions in patients already infected (N patients = 40) who were then subjected to the second dose of the vaccine, of which 13 were vaccinated with Spikevax^®^ and the remaining with Comirnaty^®^.

The elderly population (N patients = 73) was less represented in our sample, and showed a comparable distribution between men and women; they showed a slight prevalence of patients vaccinated with Comirnaty^®^ compared to other vaccines. Elderly patients mostly suffered from cardiovascular disease (N patients = 18) and metabolic comorbidity (N patients = 12), with sporadic cases of benign prostatic hyperplasia and cancer. In this elderly population, the vaccination procedure was well tolerated with an incidence of moderate ADR of 1.36% due to Comirnaty^®^. We observed a case of moderate pruritus (Table 3) in an elderly woman suffering from another pathological condition reaction already reported in the literature [55]. One hypertensive female patient with a case of persistent diarrhea and dizziness, receiving ibesartan (150 mg) and vertisec (24 mg), who was SARS-CoV-2-positive at the first dose was vaccinated with dose II of Vaxervria^®^ and doses IIII and IV of Comirnaty^®^.

Finally, a sample of pediatric patients (N patients = 58, N female = 36, male = 22) was treated mainly with Comirnaty^®^, and some with Spikevax^®^. A case of moderate vagal hypotension of non-immunological origin was detected in a female patient, something that has also been reported by others who received Comirnaty^®^ after I dose [21], and a case of redness with face swelling of allergic origin has also been observed in a male patient. In this population, the vaccination procedure was tolerated but showed a higher incidence of moderate ADR of 3.44% for Comirnaty^®^ vs. other treatments.

The calculated odds ratio of Comirnaty^®^ vs. other treatments for all age groups was 0.8, indicating a favorable outcome of this vaccine treatment vs. others.

We therefore evaluated the quality of life and the pharmaceutical service. Our data pooled on a sample of 100 patients following self-administration show that the questions regarding the quality of the pharmaceutical service highlight how the service rendered produced high satisfaction scores (Table 3), with no gender effect in the more satisfied group with code 111111, but mild problems were encountered in the group with male prevalence (N male/N female = 4/2 cases), moderate problems in the group with female prevalence (N female/N male = 14/12 cases), and severe problems and less satisfaction within the group with codes 212221, 222221, and 323331 (N female = 3 cases). Following a new interview via phone call, the patients confirmed their perceptions of the therapy and pharmaceutical service after a follow-up period of 15 months. Therefore, the scores were unmodified after 15 months of observation, and females represented the most unsatisfied patients.

The analysis (EQ-5D-Y) of the treatment prescriptions shows that 3 out of 100 patients were eligible for drug therapy, showing the maintained efficacy of the vaccine in cases of new infections on 12 March 2022. These patients were at high risk of developing severe disease; to date, they have been safely treated and not hospitalized. The long-term observation on 12 May 2023 of these high-risk patients, however, showed no hospitalization for any cause.

We therefore collected data on the COVID-19 therapy of patients under treatment in our regional area for one year, starting from May 2022 to May 2023. We found a very strong correlation of Paxlovid^TM^, distributed and dispensed by the community pharmacy in accordance with the D.P.C. protocol, with vaccine administration and the COVID-19 cases (Figure 3) in the Puglia region from August 2022 to May 2023, despite the expected lower number of patients treated with Paxlovid^TM^ vs. vaccinated patients, with 1.47% of the patients eligible for antiviral drug therapy. While an increased number of COVID-19 cases uncorrelated to vaccine administration was observed during the first four months of observations (Figure 3), the average daily numbers of hospitalizations in May and July 2022 were N = 487 and N = 497, respectively, and N = 399 and N = 401 in September and October 2023 (aggregated data), suggesting that the number of hospitalizations, despite the increase in infections, remains stable. The long-term observation of the number of hospitalizations on 12 May 2023 of all patients in our region showed no change during one year of observation.

## 4. Discussion

In our work, we evaluated the vaccine procedure in terms of tolerability and satisfaction with the pharmacy service in 550 patients in our community pharmacy. As expected, the vaccines were reactogenic after the first dose in most of the patients, with mild and moderate reactions associated with younger age and female gender as risk factors. Similarly, around 75% of the COVID-19 vaccinations led to reactogenicity, and nearly 25% of them led to one or more days of work loss, especially for patients of female gender and a younger age [56]. Uncommon reactions of a moderate–severe degree in the adult female patients were observed with Comirnaty^®^ and Spikevax^®^. The moderate immune–allergic reactions detected in our center are in line with what has been observed in the literature, wherein young women, even those without history of intolerance reactions or allergic reactions, have manifested these types of reactions to mRNA vaccines, both immediate anaphylactic and delayed, with an imbalance in favor of Comirnaty^®^ vs. Spikevax^®^ [21,55], a trend that is also confirmed in our center. In the elderly population also affected by comorbidity, the vaccination procedure was well tolerated. This could be due to the lower dose intensity protocol adopted for Comirnaty compared to that of Spikevax^®^. In the pediatric population, following the procedure of vaccination with Comirnaty^®^, an incidence of moderate ADR 3.44% higher than that observed in the adult and aged populations was observed, suggesting the higher susceptibility of these patient populations. These ADRs were syncope and immune–allergic reactions. Indeed, syncope with vagal hypotension related to anxiety of non-immunological origin in a female pediatric patient was observed in our center, and in an adult female patient with hypertension. Syncope may occur in either sex with a similar patho-mechanism due to a compensatory vagal response to the adrenergic-related anxiety, and tachycardia that leads to bradycardia and hypotension. Syncope is commonly observed with high prevalence in the female population, and hypertension is a risk factor in the adult population [57]. The gender effect of syncope can be explained by the effects of vagal innervation on the regulation of GnRH secretion and ovulation [58].

Vaccination with Comirnaty^®^ also showed a favorable O.R. < 1 vs. other treatments for all age groups. We failed to observe myocarditis in our population, although myocarditis of a moderate degree of severity was reported in 129 patients following mRNA COVID-19 vaccination, affecting mostly male patients in a metanalysis study in a large cohort of patients [19]. Also, thromboembolic events were not observed in our patients, in contrast to a recent report undertaken in a large population [20]. Despite diabetes type II and insulin response having been associated with negative prognosis in COVID-19 patients for different causes, such as associations of pathogenic factors [42,59,60], as well as loss of control and monitoring during the pandemic [61], no ADRs were observed in the diabetic patients, which represented 5.8–7% of our patients in the vaccination program. Therefore, the vaccination procedure adopted experimentally at the community pharmacy was found to be safe and effective. The population joined the vaccination campaign at the pharmaceutical site, and responded positively to the proposed pharmaceutical service. The minor reactions observed but not reported in this paper were in line with those reported in the technical data sheets of the vaccines, as were the moderate immune–allergic reactions that did not require medical intervention and hospitalization but were resolved at the patients’ homes. The trend of moderate immune–allergic reactions observed in young women in our center is in line with what has recently been reported in the literature, and supports the involvement of community pharmacies as an essential site for the systematic collection of data of epidemiological interest to direct targeted therapeutic choices. The quality of life evaluation scores were given in our center by female patients.

Aside from its role in the vaccination campaign, the community pharmacy played a key role in dispensing antivirals to the eligible patients in concert with the pharmaceutical service at the ASL-BA center responsible for the Puglia region on behalf of third parties (D.P.C.) in accordance with the protocol. It should be remembered that, in Italy, two oral antiviral drugs have so far been authorized for the treatment of COVID-19 in adults who do not require additional oxygen therapy and who have a high risk of developing a severe form of COVID-19: Paxlovid (nirmatrelvir/ritonavir) by Pfizer Europe MA EEIG and Lagevrio (molnupiravir) by Merck Sharp & Dohme.

For Paxlovid^TM^, Italy granted European authorization on 31 January 2022, as published in the Official Gazette no. 26 of 1 February 2022. This classifies the medicine for reimbursement by the National Health Service in the “C not negotiated [C(nn)]” category, and assigns the following supply regimen: medicine subject to limited medical prescription, to be renewed from time to time, and sold to the public on prescription of hospital centers identified by the regions (RNRL).

The distribution of Lagevrio^TM^ in Italy was authorized by the Decree of the Ministry of Health on 26 November 2021. Molnupiravir works by interfering with the virus’s ability to replicate; since it does not target the Spike protein, its effectiveness does not depend on the variant. Its effectiveness, however, is conditioned by the timeliness of treatment, which must begin within 5 days from the onset of disease, confirmed by positive swab, and ideally within 72 h. Progression to severe forms of COVID-19, with the involvement of the lungs, occurs after a few days from the onset of infection, but the duration of this period is very variable. The patients to be treated with molnupiravir are selected by general practitioners or Special Assistance Continuity Units (USCAs), while the prescription is the responsibility of doctors working within the facilities identified by the regions for administration. Patients who have at least one of the following risk factors associated with evolution to severe disease are eligible for treatment: active oncological or hematological diseases, chronic renal failure (excluding patients on dialysis or with eGFR < 30 mL/min/1.73 m^2^), severe broncho-pneumopathy, primary or acquired immunodeficiency, obesity (BMI ≥ 30), severe cardiovascular disease (heart failure, coronary artery disease, cardiomyopathy), and uncompensated diabetes mellitus. The therapy consists of taking four capsules a day for a total of 5 days. This drug is an important additional tool used to counteract the risks related to infection; however, it cannot be considered an alternative to vaccines. Currently, Lagevrio^TM^ is considered less cost-effective than Paxlovid^TM^.

Veklury (remdesivir) from the Gilead Sciences Company is an intravenous antiviral drug used for the treatment of COVID-19, and is also authorized in Italy. This is the first antiviral medicine to have received authorization, by decision of the European Commission of 3 July 2020, for the “treatment of coronavirus disease 2019 (COVID-19), in adult and adolescent patients (aged 12 years and older and weighing at least 40 kg) with pneumonia requiring supplemental oxygen therapy”. This authorization was implemented in Italy with the decision published in GU no. 250 of 09–10-2020, containing the classification of the medicinal product into class [C(nn)], with the following supply regimen: medicine subject to limited medical prescription, usable exclusively in a hospital environment or in a similar structure (HOSP). Since 30 December 2021, following the European authorization of an extension of indication, Veklury^TM^ has also been indicated for the treatment of COVID-19 in adults not hospitalized for COVID-19 and not underdoing oxygen therapy with the onset of symptoms for no more than 7 days, and/or presenting predisposing clinical conditions that represent risk factors for the development of severe COVID-19. 

The prescription of antivirals for the treatment of COVID-19 is subject to a monitoring register and involves the use of the card relating to the drugs subject to monitoring. In addition, all antivirals are subject to additional monitoring. This allows the rapid identification of new information. Healthcare professionals are asked to report any suspected adverse reactions using the National Pharmacovigilance Network.

Paxlovid^TM^ requires adequate training for family doctors. It provides numerous drug interactions to be evaluated carefully, but the goal is to use the drug more frequently and safely, given that, so far, its use has been lower than expected, and the current quantity may be made available to a wider audience of prescribers and patients [62]. This drug is distributed by D.P.C. in all regions, i.e., purchased by the regional ASL-BA, which makes it available in community pharmacies within 24 h of prescription. Also, in this case, the pharmacists of the community pharmacy, hospital, and territory collaborated with the national authorities to aid in the use and dispensing of the antiviral therapy, as supported by our data derived for the period 2022, after the vaccination campaign.

Our data indicate that 3% of the patient preference was for antiviral Paxlovid^TM^ after the vaccination campaign on 12 March 2022, and the vaccine’s efficacy was maintained in new infections. The percentages of patient preference, calculated based on a small sample of quality of life questionnaires, were in agreement with the trend of medical prescriptions in the entire Puglia region during one year of monitoring, starting from 1 May 2022 and extending to 1 May 2023, with 1.47% of the patients eligible for the antiviral drug therapy. During the first four months of observation, the number of COVID-19 cases was uncontrolled by the applied therapy for several reasons (for instance, the initial difficulty in patient recruitment and the unavailability of the vaccines), but a marked reduction in COVID-19 cases in our region was observed in the last 8 months of observations, this being very well correlated with therapy. These findings indicate an appropriate prescription and distribution of the therapy in our setting.

## 5. Conclusions

On behalf of third parties (D.P.C.) and in accordance with a regional protocol applied to antiviral therapy, the C.PHARM favors the capillary distribution of the drug with an approach that matches the patients’ preferred therapy and medical prescriptions, with no increase in hospitalization. The community pharmacy was demonstrated to be a safe and effective vaccination center, providing appropriate patient inclusion during the vaccination campaign, and preventing severe cardiovascular and immune-related events in high-risk patients. In this context, investigations into gender-appropriate treatment are required to overcome the higher susceptibility of female populations to the detriments of the vaccination procedure.

Community pharmacies play a crucial role as vaccination centers against the SARS-CoV-2 and influenza viruses. To reduce the pressure on the NHS during the pandemic, the government has also invited community pharmacies to participate in the NHS’s flu vaccination campaign, extending the flu vaccination network of the NHS to patients over 65, people with chronic and immunosuppressed diseases, children aged 6 months to 6 years, and pregnant or postpartum women.

The extension of the protocol to also include the herpes zoster virus, as recently proposed by Mandelli and coworkers, is under evaluation by national authorities [63].

## Figures and Tables

**Figure 1 pharmacy-12-00016-f001:**
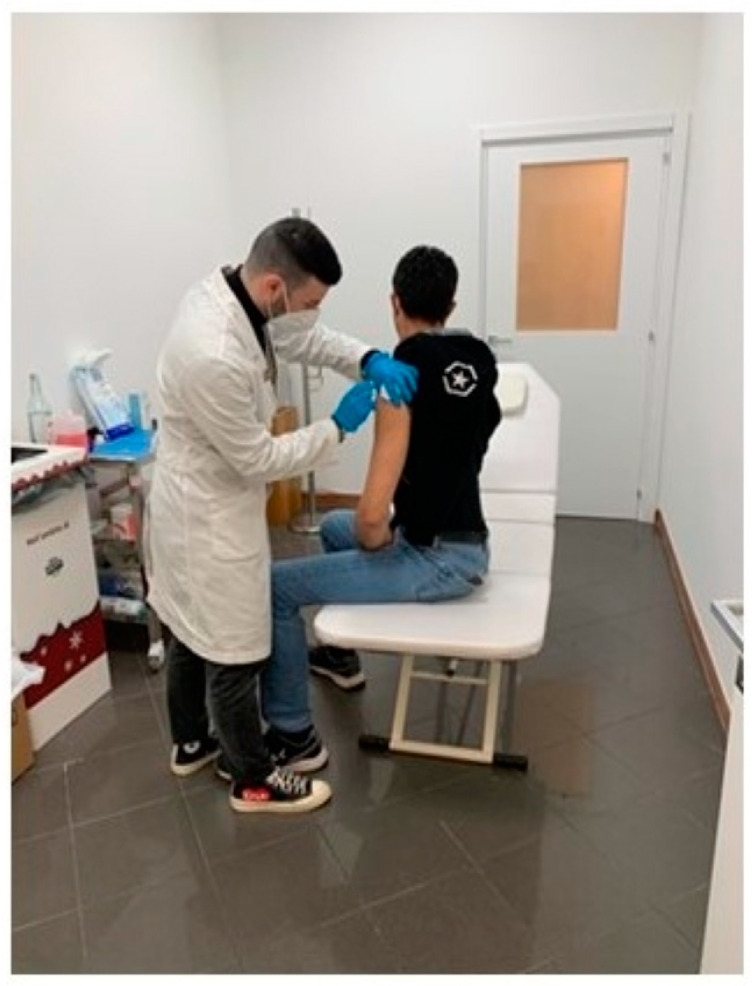
Sample image of the pharmacist (J.R.D.) who, after acquisition of written informed consent and collection of the triage form for the evaluation of the suitability/unsuitability of the patient, performed the inoculation of vaccines through the intramuscular route at an angle of 90 degrees in the deltoid muscle. To minimize injection pain, patients were invited to rotate their arm inwards, and a firm and rapid puncture was applied.

**Figure 2 pharmacy-12-00016-f002:**
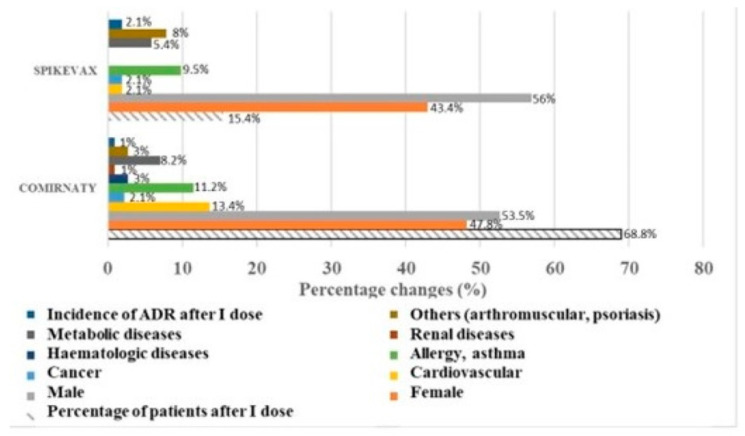
Patient distribution per gender and concomitant diseases in the two groups under treatment at the first dose; percentage incidence of moderate adverse reactions (ADRs) in adult patients (18–65 years) after vaccination related to the two vaccines administered at the vaccination center.

**Figure 3 pharmacy-12-00016-f003:**
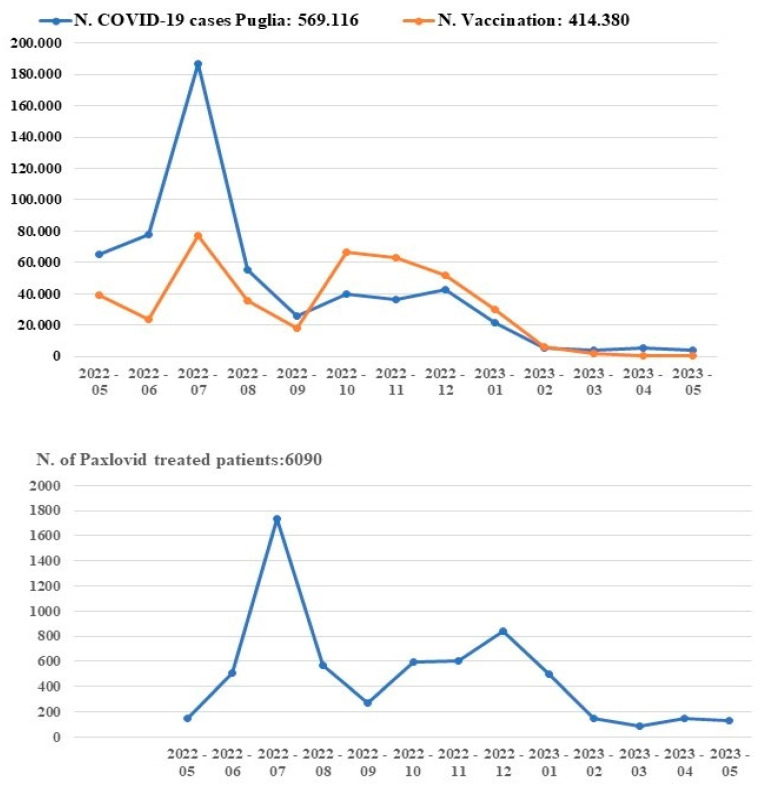
Changes in the number of COVID-19 cases with antiviral and vaccination therapy during one year of observation from 1 May 2022 to 1 May 2023 in the Puglia region. The Paxlovid^TM^ therapy was distributed on behalf of third parties (D.P.C.) in accordance with the protocol of the community pharmacy that finally dispensed the antiviral doses to the patient.

**Table 1 pharmacy-12-00016-t001:** Case reports of moderate adverse drug reactions after primary vaccination course in the adult population (18–65 years). Code: patient code; I, II, III: dose (DO) of vaccine; age (YEARS) and gender (GEN.) of the patients: F female; race and nationality (NAT.); concomitant disease and drugs of the patients; adverse drug reactions (ADRs).

Code	I DO	II DO	III Dose (Booster)	Age	GEN.	Race-NAT.	Concomitant Diseases	Drugs	ADR after I Dose
**021**	2 July 2021 COMIRNA-TY	28 July 2021 COMIRNA-TY	29 December 2021 COMIRNATY	22	F	CAUCASIAN—ITALIAN	NONE	NONE	SPASMS AND HIGH FEVER T40.5 °C
**083**	3 August 2021 SPIKEVAX	7 September 2021 SPIKEVAX	15 January 2022 SPIKEVAX	23	F	CAUCASIAN—ITALIAN	NONE	NONE	SPOTS AND REDNESS DELOCALIZED ALL OVER THE BODY
**400**	4 August 2021 COMIRNA- TY	30 August 2021 COMIRNA-TY	12 March 2022 COMIRNATY	36	F	CAUCASIAN—ITALIAN	INTOLERANCE LACTOSE, GLUTEN, NICKEL, CRUSTACEANS	NONE	EDEMA (LEGS)
**506**	15 September 2021 SPIKEVAX	14 October 2021 SPIKEVAX	11 April 2022 COMIRNATY	35	F	CAUCASIAN—ITALIAN	NONE	IRBESARTAN (150 mg), VERTISERC (24 mg)	SYNCOPE

**Table 2 pharmacy-12-00016-t002:** Moderate ADR after primary vaccination in the elderly population (>65 years) (N Comirnaty^®^ patients = 34, Vaxervria^®^ = 19, Spikevax^®^ = 11). Code: patient code; I, II, III: dose (DO) of vaccine; age (YEARS) and gender (GEN.) of the patients: F female; race and nationality (NAT.); concomitant disease and drugs of the patients; adverse drug reactions (ADRs).

Code	I DO	II DO	III Dose (Booster)	Age	GEN.	Race-NAT.	Concomitant Diseases	Drugs	ADR after I Dose
**096**	30 April 2021 COMIRNA-TY	21 May 2021 COMIRNA-TY	20 January 2022 COMIRNATY	76	F	CAUCASIAN—ITALIAN	HYPERTHYROIDISM AND ASYMPTOMATIC MYELOMA	EUTIROX	ITCHING

**Table 3 pharmacy-12-00016-t003:** Patient satisfaction after a booster dose of the vaccine. Notes: 1 (x2) = No problems, 2 = Some problems, 3 = Many problems, 9 = Missing value. 1 = No pain/discomfort; 2 = Some pain/discomfort; 3 = Considerable pain/discomfort; 9 = Missing value. 1 = Not worried/sad/unhappy; 2 = A bit worried/sad/unhappy; 3 = Very worried/sad/unhappy 1 = Very; 2 = sufficient; 3 = Little; M = Male; F = Female.

Code	Percentage Distribution of Patients (%)	Gender	Age
111111	68	M = 33, F = 35	Adult = 66
			Aged =2
111112	6	M = 4, F = 2	Adult = 6
111121	1	F = 1	Adult = 1
111131	1	M = 1	Adult = 1
111211	4	M = 3, F = 1	Adult = 4
111212	1	F = 1	Adult = 1
111221	4	F = 4	Adult = 4
111331	1	M = 1	Adult = 1
112111	2	M = 2	Adult = 2
112112	1	M = 1	Adult = 1
112211	3	M = 2, F = 1	Adult = 2
			Aged = 1
112221	5	F = 3, M = 2	Adult = 5
212221	1	F = 1	Adult = 1
222221	1	F = 1	Adult = 1
323331	1	F = 1	Adult = 1

## Data Availability

The data are available at the Pharmacy Center, and can be provided by D. Cannone.

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
