# Peer review of "The Community Pharmacy as a Study Center for the Epidemiological Analysis of the Population Vaccination against SARS-CoV-2: Evaluation of Vaccine Safety and Pharmaceutical Service"

_pharmacy, 2024, doi:10.3390/pharmacy12010016_

Round 1

Reviewer 1 Report

Comments and Suggestions for Authors

This study aimed to assess the pharmaceutical services provided at a Community Pharmacy (C.PH.) during the pandemic in the Puglia Region, focusing on vaccination outcomes, adverse reactions, service quality, and therapy preferences. An observational study was conducted on 550 patients across diverse demographics and health conditions between December 29, 2021, and March 12, 2022. Data included patient histories, post-vaccination adverse reactions, EQ5 method interviews for service evaluation, and therapy preferences. Following vaccination, reactogenicity was noted after the first dose, predominantly in younger age groups and females with mild-moderate reactions. Elderly patients exhibited better tolerance. Comirnaty displayed a favorable outcome compared to other vaccines. No observed cases of myocarditis or thromboembolic events were reported. Paxlovid was prescribed in a limited percentage of patients without impacting hospitalization rates. The Community Pharmacy served as a safe vaccination center, ensuring proper patient inclusion during vaccination. Findings highlighted vaccine reactogenicity patterns across demographics and showcased the safe distribution of Paxlovid via a novel distribution protocol.

Minor concerns:

Adverse Reaction Reporting: The description of adverse reactions appears somewhat general. Adding specific details about the types and severity of reactions would enhance the study's comprehensiveness.

Follow-up and Long-term Effects: Long-term effects of vaccinations or therapy preferences might change beyond the study period. Including a follow-up or discussing the potential for long-term effects could provide a more holistic view.

Comments on the Quality of English Language

Good.

Author Response

We thank the reviewer for her/his positive comment’s on our manuscript, the long-term data were available in May 2023 and were added to the results sections. 

Also, the syncope affecting several female patients was discussed more in detail as requested.

Reviewer 2 Report

Comments and Suggestions for Authors

I am very pleased to review this manuscript. The manuscript deals with a very important issue and contributes significantly to the communication of very important research results. 

In my view, the results and especially the figures need to be comprehensively revised. Currently, the presentation is not yet suitable for such a high-ranking journal. I recommend comprehensive revisions here. 

Furthermore, the discussion is far too superficial. It needs to be deepened and made more scientific. 

To summarize, I strongly recommend major revisions. These should be made as soon as possible, as the topic is of very high relevance to the scientific and medical community worldwide.

Comments on the Quality of English Language

Minor editing of English language required

Author Response

We thank the reviewer for her/his positive comments on our manuscript, the results and figures were fully revised. The discussion was also revised discussing the pathophysiological basis of syncope in females.

All manuscript was English edited by mdpi service.  

Reviewer 3 Report

Comments and Suggestions for Authors

The paper requires a complete review and edit of the English.  

Comirnaty need a registration mark after it (R)

Pavlovid is a trademark name TM

In many locations the words are hyphenated  like at line 142 re-view should be review.  This type of occurrence is through-out the paper.

100 thousand - should be one-hundred thousand line 182.  This is one example of the need for English adjustment.  

Line 203 - is another example of inadequate English.  The communication is of concern.  This makes no sense.  There are many sentences and related that make no sense. 

line 414 - be paid to.  You are not providing someone with money.  A careful read of the English in this paper is needed. 

line 706 - rush should be rash.  Again English.  There are numerous issues of this nature in the paper. 

Fig 3 - what is the red line.  no description. 

A complete overhaul of the English and a detailed read of the paper for structure and clearly is required.  The paper can also be shortened.  I suggest a re-write of this paper be conducted. 

The paper is worth publishing with major English adjustment and the manuscript being shortened. 

Comments on the Quality of English Language

A complete re-write of the paper's English is needed.  Many places the language makes no sense.  A native English speaker is needed to go over the paper.  I also suggest making the paper shorter.  

Author Response

The paper requires a complete review and editing of the English.  

Ok

Comirnaty needs a registration mark after it (R)

OK

Pavlovid is a trademark name TM OK

In many locations the words are hyphenated like in line 142 re-view should be reviewed.  This type of occurrence is throughout the paper.

OK revised

100 thousand - should be one-hundred thousand line 182.  This is one example of the need for English adjustment.  

OK

Line 203 - is another example of inadequate English.  The communication is of concern.  This makes no sense.  There are many sentences and related ones that make no sense. 

OK

line 414 - be paid to.  You are not providing someone with money.  A careful read of the English in this paper is needed. 

OK

line 706 - rush should be rash.  Again English.  There are numerous issues of this nature in the paper. 

Fig 3 - what is the red line.  no description. 

All figures were revised

A complete overhaul of the English and a detailed read of the paper for structure and clarity is required.  The paper can also be shortened.  I suggest a rewrite of this paper be conducted. 

The paper is worth publishing with major English adjustments and the manuscript being shortened. 

OK, the mdpi service provided the new English editing

Round 2

Reviewer 2 Report

Comments and Suggestions for Authors

The revisions made to the manuscript are truly impressive, showing meticulous application of all the recommended changes. This has significantly elevated the work's quality and scientific value. The thoroughness and attention to detail stand out remarkably.

The revised manuscript not only reflects a deep understanding of the subject but also a strong commitment to academic excellence. The integration of feedback has enhanced the clarity and coherence of the work, positioning it as a substantial contribution to the field.

Given its high scientific merit, there is a strong recommendation for the manuscript to be published as quickly as possible. The exceptional effort and dedication put into improving it have certainly paid off, and it's expected that the research will be well-received in the academic community.